# An integrated chromatin accessibility and transcriptome landscape of human pre-implantation embryos

Longqi Liu[1,2], Lizhi Leng[3,4], Chuanyu Liu[1,2,5], Changfu Lu[3,4,6], Yue Yuan[1,2,5], Liang Wu[1,2,5], Fei Gong[3,4,6], Shuoping Zhang[3,4], Xiaoyu Wei[1,2,5], Mingyue Wang[1,2,5], Lei Zhao[6], Liang Hu[3,4,6,7], Jian Wang[1,8], Huanming Yang[1,8], Shida Zhu[1,2], Fang Chen[1,2,9], Guangxiu Lu[3,4,6,7], Zhouchun Shang [1,2,10] & Ge Lin[3,4,6,7]

Human pre-implantation embryonic development involves extensive changes in chromatin structure and transcriptional activity. Here, we report on LiCAT-seq, a technique that enables simultaneous profiling of chromatin accessibility and gene expression with ultra-low input of cells, and map the chromatin accessibility and transcriptome landscapes for human pre-implantation embryos. We observed global difference in chromatin accessibility between sperm and all stages of embryos, finding that the accessible regions in sperm tend to occur in gene-poor genomic regions. Integrative analyses between the two datasets reveals strong association between the establishment of accessible chromatin and embryonic genome activation (EGA), and uncovers transcription factors and endogenous retrovirus (ERVs) specific to EGA. In particular, a large proportion of the early activated genes and ERVs are bound by DUX4 and become accessible as early as the 2- to 4-cell stages. Our results thus offer mechanistic insights into the molecular events inherent to human pre-implantation development.

[1] BGI-Shenzhen, Shenzhen 518083, China. [2] China National GeneBank, BGI-Shenzhen, Shenzhen 518120, China. [3] Institute of Reproductive and Stem Cell Engineering, School of Basic Medical Science, Central South University, Changsha 410078, China. [4] Key Laboratory of Stem Cells and Reproductive Engineering, Ministry of Health, Changsha 410078, China. [5] BGI Education Center, University of Chinese Academy of Sciences, Shenzhen 518083, China. [6] Reproductive & Genetic Hospital of CITIC-Xiangya, Changsha 410008, China. [7] National Engineering and Research Center of Human Stem Cell, Changsha 410078, China. [8] James D. Watson Institute of Genome Sciences, Hangzhou 310013, China. [9] Laboratory of Genomics and Molecular Biomedicine, Department of Biology, University of Copenhagen, Copenhagen 2100, Denmark. [10] Department of Regenerative Medicine, Tongji University School of Medicine, Shanghai 200092, China. These authors contributed equally: Longqi Liu, Lizhi Leng, Chuanyu Liu, Changfu Lu, and Yue Yuan. Correspondence and requests for materials should be addressed to Z.S. (email: shangzhouchun@genomics.cn) or to G.L. (email: linggf@hotmail.com)

Early mammalian embryos undergo widespread epigenetic reprogramming to allow the conversion of terminally committed gametes to a totipotent state[1]. It is therefore of crucial importance to map the chromatin state of regulatory elements and the transcriptional outcomes using omics tools during this process to understand the role of major *cis*-regulatory elements (e.g., promoters and enhancers) or *trans*-factors (e.g., transcription factors (TFs) and epigenetic modifiers) that drive embryonic development. By using mouse models, previous studies have demonstrated multiple landscapes including the transcriptome[2,3], methylome[4,5], chromatin accessibility[6,7], histone modifications[8–10], and 3-D genome contacts[11,12]; which can precisely characterize the key molecular events during mammalian embryo development. However, because of the limitations on low-input technologies and the difficulties in obtaining human embryonic materials, the dynamics of higher-order chromatin structure (e.g., chromatin accessibility (CA) and histone modifications) in early human embryogenesis remain poorly understood.

In this study, we develop LiCAT-seq (low-input chromatin accessibility (CA) and transcriptome sequencing), a technique that enables the simultaneous assay of CA and gene expression (GE) with low-input samples (Fig. 1a). We apply this technique to profiling chromatin structure and GE dynamics during human pre-implantation embryos and demonstrate the key regulatory dynamics for genes activated during embryonic genome activation (EGA). In particular, we find that a large proportion of the early activated genes and endogenous retrovirus (ERV) elements possess DUX4-binding sites and become accessible as early as the 2- to 4-cell stages. In one such example, we observe widespread DUX4 binding on MLT2A1, which flanks HERVL, a subfamily of ERV that become accessible during major EGA. Our work thus suggests the important roles of early TFs in the remodeling of the closed chromatin during human pre-implantation embryo development.

## Results

**Profiling of CA and GE with low-input samples** . LiCAT-seq physically separates cytoplasm and nuclei, enabling parallel library construction for CA and GE profiles from both cellular components. The cytoplasm containing mRNA was subjected to a modified Smart-seq2[13] protocol (Fig. 1a and Methods); whereas for ATAC-seq libraries of the nuclei, we made some modifications to the conventional ATAC-seq protocol[14] to reduce the loss of low-abundant genomic DNA. The major improvements included: (1) complete lysis of nuclei after a Tn5 tagmentation step; and (2) purification of genomic DNA after pre-amplification using primers targeting Tn5 adaptors. To validate LiCAT-seq, we first applied this integrated approach to both human embryonic stem cells (hESCs) and hESC-derived hepatocyte-like cells (see Methods). We found that our LiCAT-seq profiles generated from as few as 10 cells could recapitulate results generated from bulk (50,000) cells. For example, LiCAT-seq-generated CA data showing a high enrichment of reads around transcription start site (TSS) regions—and the correlations between profiles generated from 10 cells and bulk cells were high (Supplementary Figure 1a, b). Interestingly, when promoters were categorized based upon high, intermediate and low-CpG content (high-CpG-density promoters (HCPs), intermediate-CpG-density promoters (ICPs), and low-CpG-density promoters (LCPs)), we observed a stronger enrichment of CA reads at promoters with a higher GC ratio, which is similar to the enrichment of histone H3 lysine 4 trimethylation (H3K4me3)[15], suggesting a potential synergistic function of CA and H3K4me3 (Supplementary Figure 1c). The enrichment of CA reads in high-GC regions is not likely owing to

technical bias (e.g., bias from Tn5 and DNA polymerase), because we observed a significantly higher enrichment of LiCAT-seq signal on known DNase I-hyposensitive sites than other sites with a similar level of GC content (Supplementary Figure 1c). In addition, LiCAT-seq-generated GE data showed strong reproducibility and robustness in the capture of mRNA transcripts (Supplementary Figure 1d, e). Moreover, comparison of both omics in these two cell types validated the ability of LiCAT-seq in the detection of major events during ESC differentiation, such as decreased expression of the pluripotency genes *OCT4* and *NANOG* (Supplementary Figure 1f, h), as well as the reduced accessibility to OCT4- and NANOG-binding sites[16] (Supplementary Figure 1g, h). We also applied LiCAT-seq to two stages of mouse embryos (4-cell and morula stages) (Methods, Supplementary Figure 1, 2), and observed both high reproducibility and successful identification of early events, including the activation of *OCT4*[6,7] (Supplementary Figure 2). Collectively, our results indicate that LiCAT-seq enables precise measurement of CA and GE dynamics—even with limited input materials.

**CA and GE dynamics during early embryonic development**. To understand how the chromatin regulatory landscape is established during human pre-implantation development, we applied the LiCAT-seq approach to human embryos from the zygote to the blastocyst stages (with separated inner cell mass [ICM] and trophectoderm [TE]); metaphase II (MII) oocytes and sperm were included as well (Supplementary Figure 3a). Between 9 and 17 embryonic cells were collected and two biological replicates were included at each developmental stage (Supplementary Table 1 and Supplementary Data 1). We then obtained CA and GE profiles for all of the stages—except for GE data sets from sperm, which were excluded because they showed a very low rate of mapping to the genome. The data in the independent biological replicates were highly comparable (Supplementary Figure 3b, c); and as expected, we observed a strong read enrichment around TSSs and within HCP regions in CA profiles (Supplementary Figure 3d, e), as well as a large number of genes at all of the stages in GE profiles (Supplementary Figure 3f).

We observed a progressive increase in accessible regions from 2-cell to later embryonic stages (Fig. 1b), which is in agreement with previous investigations of mouse embryo development[6,7]. Notably, although the oocyte chromatin is largely inaccessible (with only 54 peaks detected in both replicates), the number of accessible regions in sperm is relatively high (with 10,385 peaks detected in both replicates). Interestingly, we observed a transient opening of chromatin at the zygote stage (Fig. 1b and Supplementary Figure 4a), which is likely to be reasonable given the accessible features of sperm chromatin (Fig. 1b and Supplementary Figure 4b). To validate this, we compared the accessible regions between sperm and zygote. Surprisingly, the CA reads along the sperm genome were poorly enriched in regions that were open in the zygote (Supplementary Figure 4c), and these were extensively distributed on gene-poor regions, exhibiting a pattern opposite to all other stages including the zygote (Fig. 1d). To further quantify this, we binned the genome into 2 Mb windows and plotted CA read density versus the number of genes within each window. As expected, in the oocyte and at embryonic stages, the number of reads for a given window was positively correlated with the number of genes in that window, whereas in sperm we observed a strong negative correlation (Fig. 1e). These results are supported by an earlier study showing preferential nucleosome retention at gene deserts in sperm relative to other cell types[17]. This unique feature of sperm chromatin in gene-poor regions might be important for the access of maternal TFs, enabling complete reprogramming of

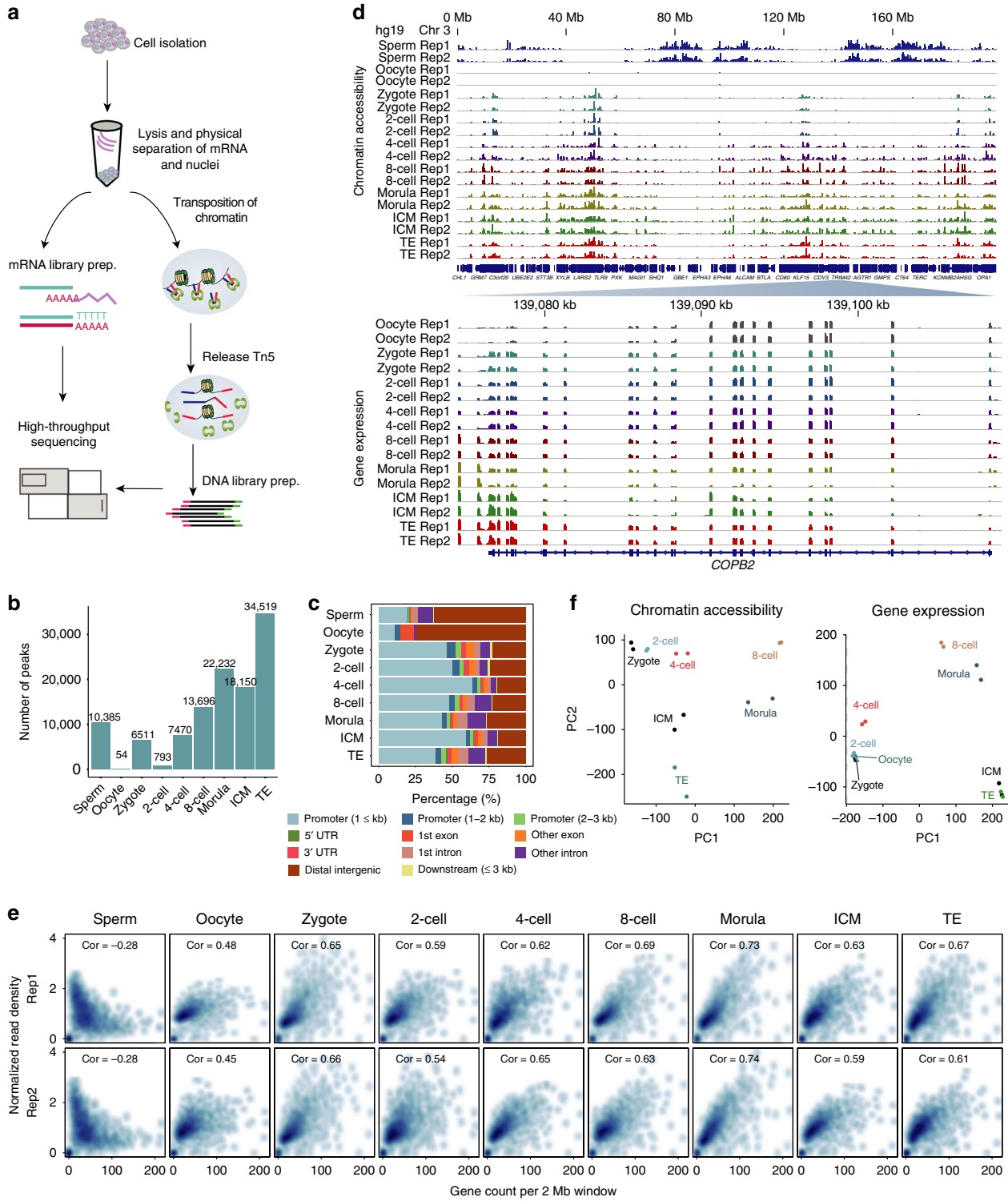

**Fig. 1** The accessible chromatin landscapes of human pre-implantation embryos. **a** Schematic representation of LiCAT-seq for simultaneous profiling of chromatin accessibility and transcriptome with ultra-low-input of cells. **b** Number of accessible regions detected at the indicated stages of embryo development. **c** Genomic distribution of chromatin accessibility peaks detected at the indicated stages. **d** Genome browser view of a representative region showing the chromatin accessibility and gene expression signals at each developmental stage. **e** Scatterplot of gene density (*x* axis) versus normalized read density (*y* axis) at each developmental stage. **f** Principal component plots of normalized chromatin accessibility and gene expression signals

the paternal genome, where the gene-rich regions are largely protected by protamine. Interestingly, a survey of TF motifs on the accessible regions of the zygote genome identified several

motifs with significant enrichment (Supplementary Figure 4d). Most of the TFs with top *p* values also exhibited high expression levels at this stage, including *SP1*, *KLF5*, and *NFYA*

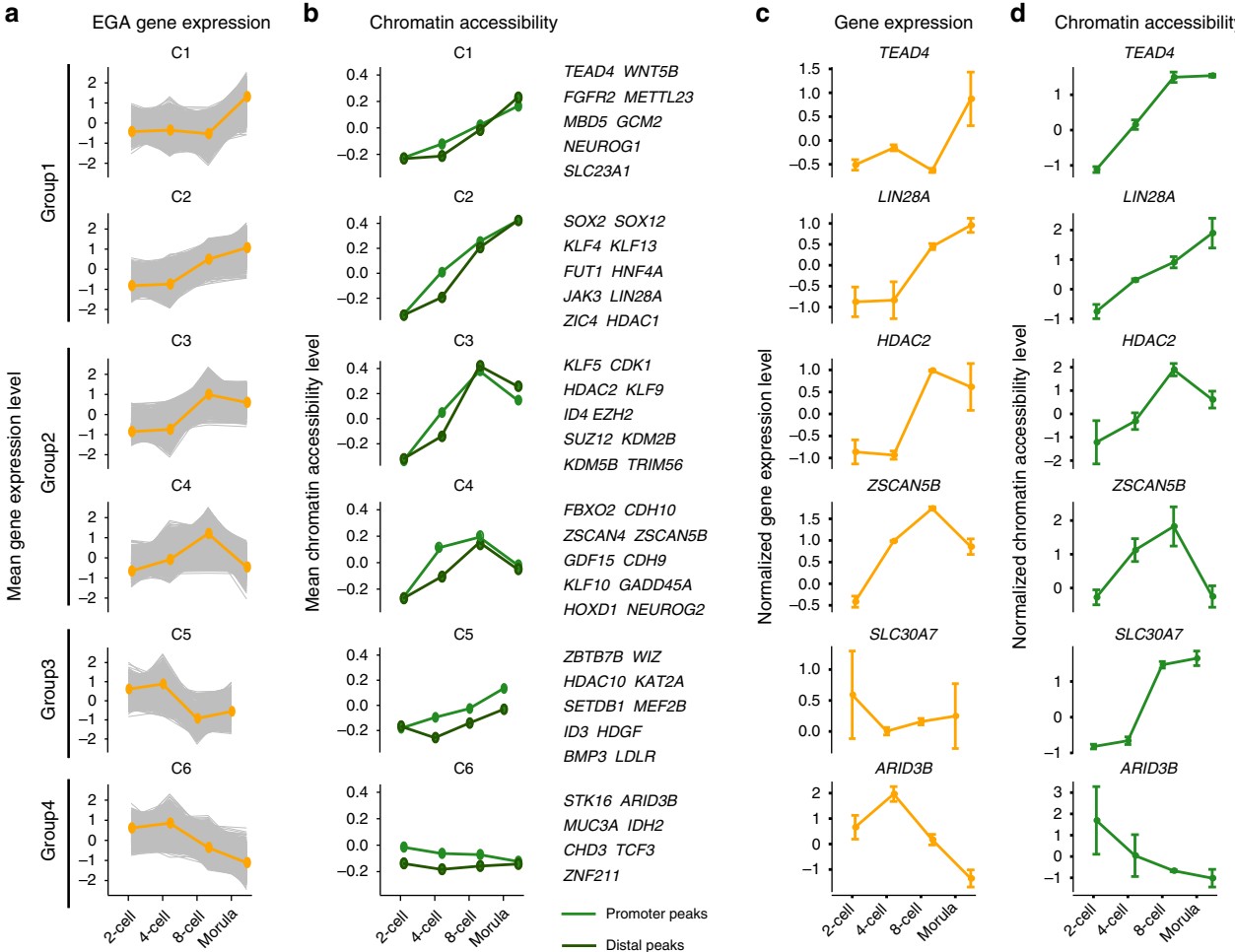

**Fig. 2** Decomposition of gene expression and chromatin accessibility dynamics during embryonic genome activation. **a** Fuzzy clustering analysis of EGA gene expression signals for the indicated stages. The individual gray lines represent the expression level of individual genes, and the orange line represents the value for the cluster center. **b** Mean standardized chromatin accessibility of promoter peaks (light green) and distal peaks (dark green) for EGA genes in each cluster in **a**. Example genes for each cluster are listed in the right panel. **c**, **d** The dynamics of gene expression **c** and chromatin accessibility **d** for representative genes in each cluster. Source data used in this figure are provided as Supplementary Data 2. The error bars shown in this figure represent the mean ± standard deviation (SD) of two replicates

(Supplementary Figure 4e), suggesting strong transcriptional activity. Collectively, our results suggest that the presence of maternal TFs—rather than paternal genome accessibility—might provide a possible explanation for the transient opening of the zygote genome.

Principal component analysis (PCA) of CA and GE data showed similar degrees of discrimination for different developmental stages of embryos. For example, both datasets showed minor changes before the 2-cell stage, but striking changes in subsequent stages (Fig. 1f), suggesting synergistic regulation of chromatin structure and GE during pre-implantation embryo development.

**Accessible chromatin is associated with EGA.** EGA is one of the most important events that takes place during human pre-implantation development; embryos progress from a state of transcriptional quiescence to a state where potentially thousands of genes are transcribed. To comprehensively study this process, we focused our analysis on EGA and investigated the impact of the establishment of accessible chromatin on transcriptional activity during this process. We first applied the Mfuzz clustering method[18] to CA profiles from 2-cell to morula-stage embryos,

yielding six clusters of regulatory regions that showed distinct dynamic patterns (Supplementary Figure 5a). Genomic regions in most of the clusters (C1–5) showed increasing accessibility and were enriched at sites distal ( > 5 kb) to the TSSs of genes, whereas regions that showed increased accessibility at both the 4- and 8-cell stages were primarily enriched at promoter-proximal regions ( < 1 kb) (Supplementary Figure 5b), suggesting distinct roles for proximal and distal elements during EGA. To gain insights into the impact of the widespread rewiring of CA on the activation of genes, we investigated the putative transcriptional targets (including proximally and distally regulated genes) of each cluster. Notably, we observed a prevalent decrease in the expression of genes associated with each CA cluster (Supplementary Figure 5c), which is not surprising considering the widespread degradation of maternally inherited transcripts[1].

To explore the transcriptional dynamics in EGA that occur without the influence of maternal transcripts, we focused principally on genes that were up-regulated during EGA (denoted hereafter as "EGA genes", Supplementary Data 2). When we clustered EGA genes into six classes by the Mfuzz method[18] and investigated the accessibility levels of both promoter-proximal and -distal regulatory regions, we observed synchronous dynamics of CA and GE (Fig. 2a, b), suggesting an important

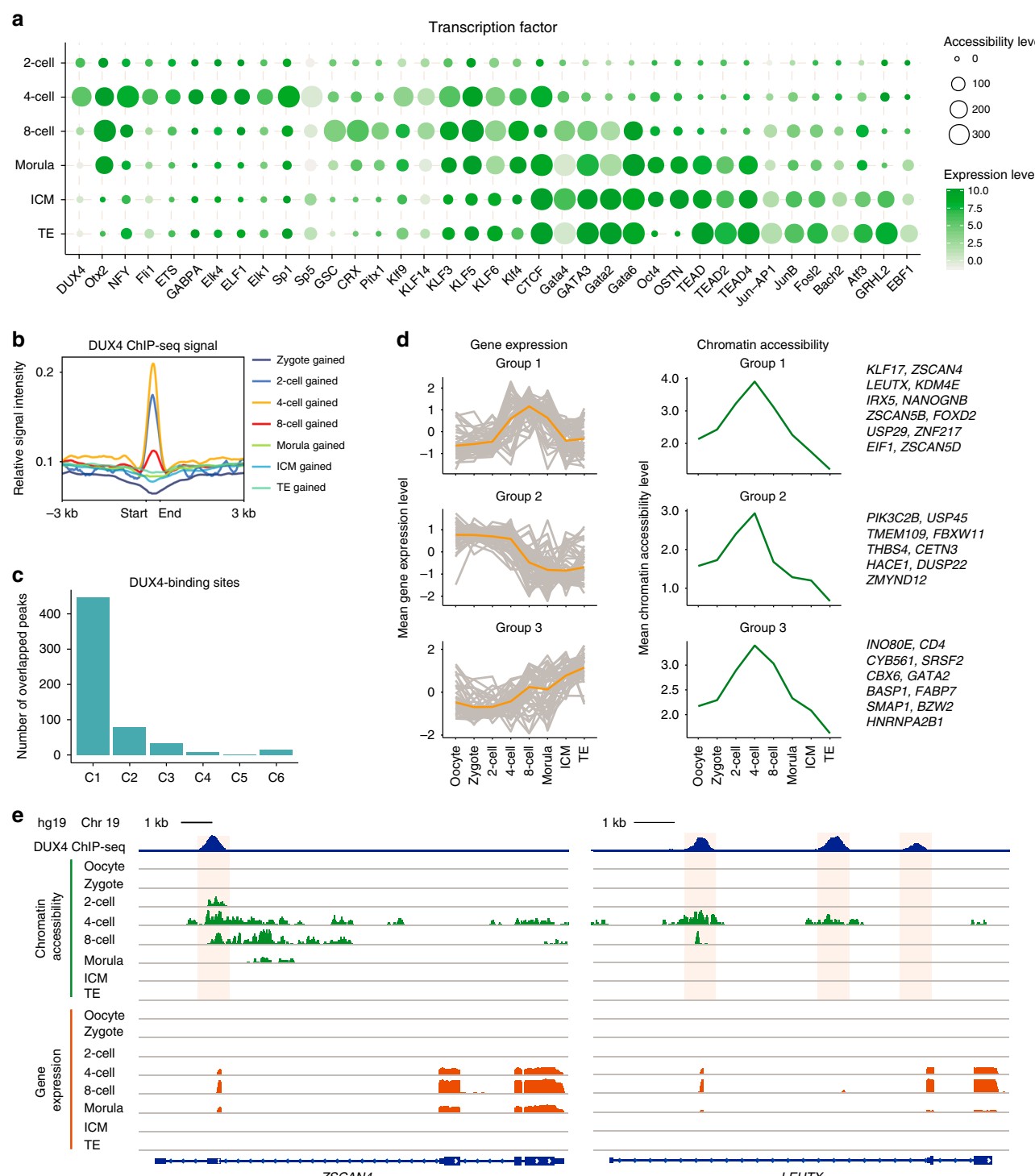

**Fig. 3** Identification of key transcription factors during human pre-implantation development. **a** Enrichment of TF motifs within the gained accessible regions at the indicated stages. Each point represents a significant enrichment for the indicated motifs. The point size represents the motif-enrichment *P* value (−log *P* value) and the color represents gene expression level of the corresponding TFs. OSTN represents OCT4-SOX2-TCF-NANOG, and the expression level for OSTN is the mean expression level of the four genes. **b** Enrichment of DUX4 ChIP-seq signals in accessible regions gained at each development stage. **c** Number of DUX4-binding sites that overlap with the accessible regions in the six chromatin accessibility clusters in Fig. 2a. **d** Left panel: fuzzy clustering analysis of DUX4 target gene expression level at the indicated stages. The individual gray lines represent the expression level of individual genes, and the orange line represents the value for the cluster center. Middle panel: dynamics of mean values of normalized chromatin accessibility signals for all regions corresponding to the genes analyzed in the left panel. The example genes are listed in the right panel. **e** Genome browser views showing DUX4 binding, chromatin accessibility, and gene expression signal around *ZSCAN4* (left panel) and *LEUTX* (right panel). Source data used in this figure are provided as Supplementary Data 3

role for chromatin structure remodeling in the activation of EGA genes. Interestingly, the distal regions showed a slower response than promoter-proximal regions at the 4-cell stage, whereas at later stages we observed a similar level of accessibility at both regions (Fig. 2b). This suggests that promoters that opened at the 4-cell stage may cause distal regions to open at later stages. To investigate this potential relationship, we uncovered four distinct groups of genes defined by the regulatory patterns. The first group consisted of genes activated at later stages (8-cell or morula), where these genes became accessible early and remained open to the morula stage (C1 and C2), suggesting potential roles at later stages. Many known pluripotency or developmental genes, such as SOX2 and TEAD4, are included in this group (Fig. 2b–d). The second group consisted of genes that were activated transiently at the 4- or 8-cell stage and exhibited reduced expression at the morula stage (C3 and C4). These genes gained accessibility quickly and reverted after the 8-cell stage, implying roles that are restricted to the early stages. Supporting this is the inclusion of genes (such as ZSCAN4, KDM4E, and KLF17) that have been reported to be transcriptionally activated during human or mouse pre-implantation development (Fig. 2b–d)[19]. The third group contained genes that were downregulated early, but reactivated at the morula stage (C5). Surprisingly, this group of genes exhibited continuous up regulation with respect to accessibility, likely owing to a regulatory mechanism that enables the replacement of maternal transcripts by transcription of the same genes from the embryonic genome (Fig. 2a–d). The final group was comprised of genes that exhibited a transient increase in expression, followed by a rapid decrease as the embryo approached the morula stage (C6). These genes showed a loss of accessibility during the entire EGA process (Fig. 2a–d). The short activation of these genes may be owing to post-transcriptional regulation; however, because of the closed state of the local chromatin, the expression could not be continued during later stages. Collectively, our analyses revealed a strong relationship between the establishment of accessible chromatin and EGA gene activation.

It is worth noting that a previous study on DNA methylome dynamics of human pre-implantation embryos showed dramatic de novo DNA methylation during the 4- to 8-cell stage transition[20]. While investigating the CA and GE dynamics of DNA methyltransferases (DNMT1, DNMT3A, DNMT3B, and DNMT3L) and demethylases (TET1, TET2, and TET3), we observed quite similar dynamics on these two omics layers (Supplementary Figure 5f). Intriguingly, the de novo DNA methyltransferase DNMT3L showed a dramatic increase in both CA and GE, whereas the other enzymes from the same family showed an opposite tendency. This suggests a potential role of DNMT3L in the regulation of de novo DNA methylation during EGA. Notably, we observed that within the DNA demethylases, the expression patterns were rather different. For example, TET3 exhibited reduced expression throughout the entire process, a pattern supported by an earlier study that showed a specific role for TET3 DNA dioxygenase in epigenetic reprogramming by oocytes[21]. In contrast, TET1 and TET2 were upregulated during EGA, suggesting that the transition from DNA methylation to 5-hydroxymethylcytosine might occur during this process.

**TF binding dynamics during embryonic development**. Regulatory regions (such as promoters or enhancers) are comprised of clusters of TF motifs that allow the binding of master regulators that drive gene transcription[22]. To comprehensively identify the key TFs at each stage, we focused on the regions that became accessible at different stages (Supplementary Figure 6a, b), and computed the enrichment of TFs using HOMER[23]. We found a series of TFs that showed specific enrichment at different stages (Fig. 3a). Interestingly, we observed high enrichment of the pluripotency factors OCT4 or OCT4-SOX2-TCF-NANOG at both the morula stage and in ICM cells, which is in accordance with the observation that OCT4 is required for the maintenance of pluripotency[24]. We also found specific enrichment of the JUN/FOS motif in TE cells, but not in ICM cells, supporting a role for JUN/FOS TFs (e.g., JUNB and FOSL2) in the regulation of trophoblast-specific gene expression[25]. We also found that GSC and CRX appeared to play a specific role at the 8-cell stage and these genes have previously been shown to be markers of lineage specification[26,27], suggesting that the differentiation program may be initiated at this stage. Notably, some of the TFs known to be important in mouse development—including Nr5a2 and Rarg—are not enriched in the same stages in human development (Supplementary Data 3), suggesting a species-specific regulatory mechanism.

Recent findings have revealed that double homeobox 4 (DUX4) plays a key role in regulating the EGA process in placental mammals[19,28,29]. However, owing to the lack of epigenomic datasets of human pre-implantation embryos, the mechanism by which DUX4 regulates its target genes remains poorly understood. Consistent with previous findings using RNA-seq, our GE data showed transient activation of DUX4 from the zygote to 4-cell stages (Supplementary Figure 6c). We also observed that a high enrichment of DUX4, which together with NFY and SP1, has also been reported to be an early regulator in mouse embryos[7] at the 4-cell stage, followed by a rapid loss of enrichment at later stages (Fig. 3a). To understand the relationship between DUX4 binding and the dynamics of accessible chromatin, we integrated published DUX4 chromatin immunoprecipitation followed by sequencing (ChIP-seq) data[19] with the regions that became accessible at each stage. We observed that although DUX4 ChIP-seq signals were enriched at the 2- and 4-cell stages, they were substantially reduced at the 8-cell or later stages (Fig. 3b). To study the accessibility of the direct DUX4-binding sites, we calculated the intersection between DUX4-binding sites (as detected by ChIP-seq) detected in embryonic stem cells (ESCs) and regions that became accessible at the 2- and 4-cell stages. We found a strong overlap of accessible regions, with 697 out of 2782 DUX4-binding sites open at the early stages (Supplementary Figure 6d). Concomitantly, we found a large overlap between DUX4-binding sites and the C1 cluster of accessible regions (Fig. 2a and Fig. 3c), supporting an important role for DUX4 in the initiation of EGA during early embryo development.

Next, we examined the expression of DUX4 target genes, which were defined as genes that are located 10 kb up- or downstream of DUX4-binding sites. We noted that although DUX4-binding sites exhibited transient opening of chromatin between the 2- and 8-cell stages, the expression levels of many DUX4 target genes changed asynchronously. These expression changes showed three different patterns and we grouped the genes accordingly (Fig. 3d and Supplementary Figure 6e). The first group of genes transiently increased at the 4- and 8-cell stages. Notably, genes within this group included key regulators that are activated during overexpression of DUX4 in iPSC or mESCs[19,28,29] such as ZSCAN4, LEUTX, KDM4E, and KLF17 (Fig. 3d, e and Supplementary Figure 6f). The second group of genes showed progressive downregulation during embryonic development, suggesting that they may have originally been maternal transcripts (Fig. 3d and Supplementary Figure 6f). In contrast, genes in the third group showed progressively increasing expression levels, which suggests that DUX4 may initiate the binding of other TFs that play crucial roles in the late stages of embryonic development (Fig. 3d and Supplementary Figure 6f). Overall, by integrating CA and GE data, our analyses suggest that

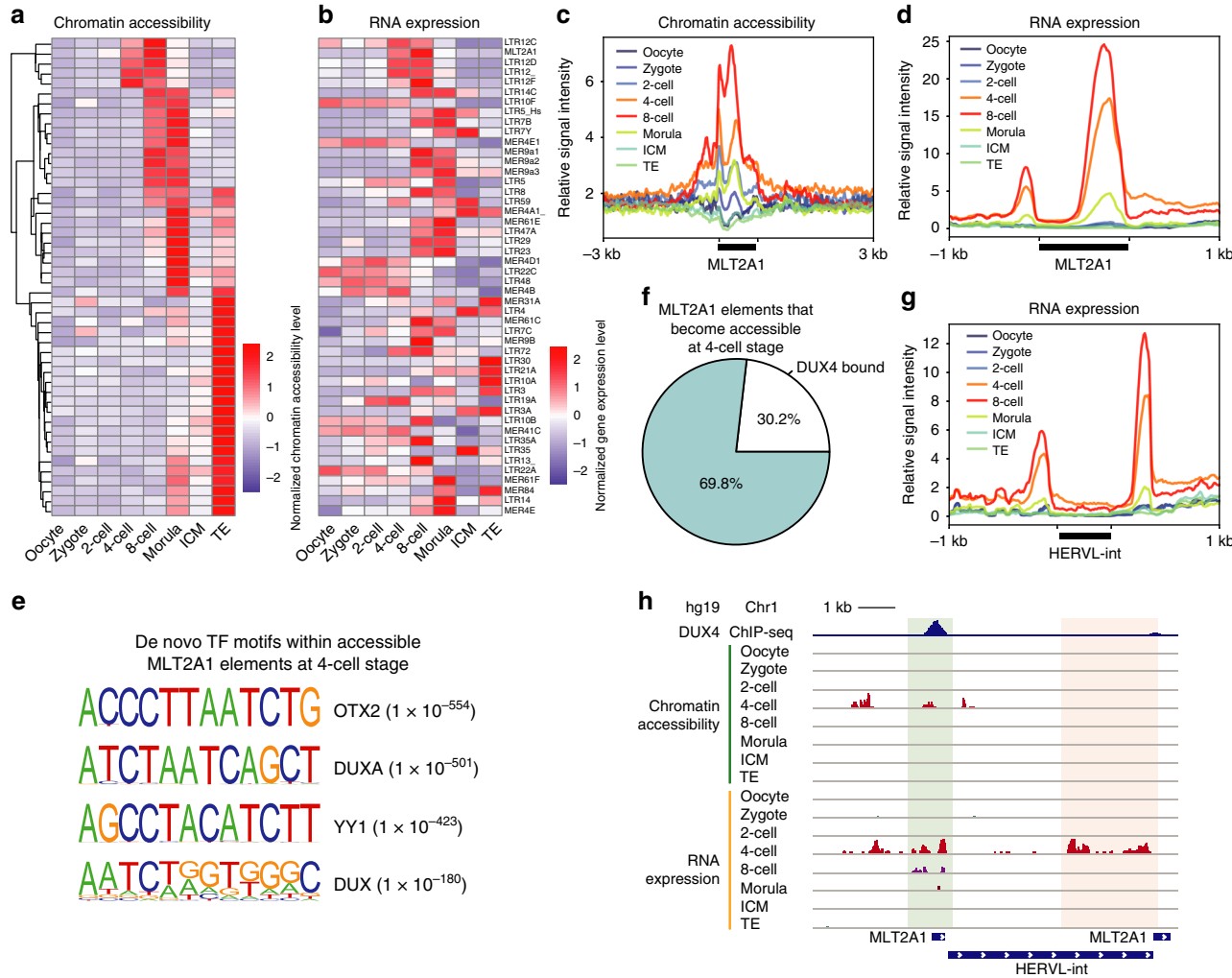

**Fig. 4** Chromatin accessibility and expression dynamics of retrotransposons during human pre-implantation development. **a**, **b** Heat maps showing chromatin accessibility **a** and expression **b** levels of retrotransposons at the indicated development stages. **c**, **d** Chromatin accessibility **c** and RNA **d** read enrichment around MLT2A1 elements at the indicated stages. **e** Enrichment of the indicated transcription factor motifs found within MLT2A1 elements that become accessible at the 4-cell stage. **f** Proportion of MLT2A1 elements that are bound by DUX4 within all MLT2A1 elements that become accessible at the 4-cell stage. **g** RNA read enrichment around HERVL elements at the indicated stages. **h** Genome browser views showing DUX4 binding, chromatin accessibility, and gene expression signal around a HERVL element flanked by MLT2A1. Source data used in this figure are provided as Supplementary Data 4

DUX4 triggers EGA gene activation by inducing extensive chromatin remodeling at early stages.

**CA and expression dynamics of retrotransposons**. Retrotransposons are remnants of ancient viral infections that account for over 40% of the human genome[30], and include long terminal repeats containing elements (termed "LTR retrotransposons" or "endogenous retroviruses (ERVs)") and LTR-lacking elements (such as LINEs, SINEs, and SVA elements)[30]. Although it has been shown that LTR elements can function as regulatory elements and possibly influence the transcription of host genes[31], a systematic survey of the CA landscape of all of the retrotransposons during human pre-implantation development is still lacking. To explore this systematically, we analyzed dynamics of all of the known families of retrotransposon elements using normalized CA and GE reads at each stage (Supplementary Data 4), and we found that almost all of the top specific retrotransposons belong to the ERV/LTR family. Interestingly, the earliest opening of ERV elements appeared at the 4-cell stage (e.g.,

LTR12C and MLT2A1). These ERVs maintained accessibility at the 8-cell stage, but then showed a loss of accessibility at the morula stage (Fig. 4a, b and Supplementary Figure 7a, b). In addition, we also found that LTR families, including LTR5_Hs and LTR7B, became accessible at the 8-cell stage but exhibited a loss of accessibility after the morula stage (Fig. 4a, b and Supplementary Figures. 7a, b and 8a, b). This is consistent with the transcriptional levels of LTR5_Hs and LTR7B and the corresponding HERVK and HERVH elements, which have been reported to be essential to the induction of viral restriction pathways and establishment of naive pluripotency[32,33], respectively. Furthermore, we observed high enrichment levels of ERV elements (such as LTR21A and LTR3) in TE cells, suggesting that ERVs play a role in the regulation of TE differentiation (Fig. 4a, b and Supplementary Figure 7a, b). Collectively, our analyses revealed a strong contribution of chromatin reconfiguration in the activation of ERVs during embryogenesis. Notably, although most of the ERVs were significantly enriched at specific stages (4-cell to TE of the blastocyst stage), some of the ERVs still showed asynchronous dynamics on accessibility and expression; for

example, many of them (such as LTR10F and LTR22C) showed their highest level of expression in the oocyte and exhibited reduced expression at later stages (Fig. 4b), reflecting transcripts of maternal origin. Although these elements underwent transient opening at the 8-cell or morula stage, they were not transcriptionally reactivated (Fig. 4a), suggesting that other factors (such as DNA methylation) participate in the repression of ERV elements.

LTR elements have been shown to be activated by transcription factors. For example, LTR5_Hs is bound by OCT4, a master pluripotency regulator, during the late stages of pre-implantation embryo development[32]. Although a series of TFs were identified as early regulators, the exact TFs that activate ERVs during EGA remain poorly understood. We next asked whether the combined CA and GE datasets could be helpful in precise identification of the key TFs. We focused on the LTR12C and MLT2A1 ERV families, which showed stage-specific expression levels and synergistic dynamics in both the epigenome and transcriptome at the 4- and 8-cell stages (Fig. 4c, d and Supplementary Figure 8c, d). When we computed the enrichment of TF motifs using HOMER[23], we found that the early regulators, such as NFYB and ZSCAN4, showed high enrichment on LTR12C (Fig. 4e and Supplementary Figure 8e), suggesting a potentially intriguing role for TFs in the activation of early ERVs. Surprisingly, we observed a significant enrichment of OTX4 and DUX family TFs on MLT2A1, but not on LTR12C (Fig. 4e and Supplementary Figure 8e). To further investigate this observation, we calculated the intersection of binding sites for OTX4 and DUX4 in human ESCs and the accessible MLT2A1 elements at the 4-cell stage. We found that 30.2% of the MLT2A1 elements that were accessible at the 4-cell stage were bound by DUX4 (Fig. 4f), whereas only 9.5% of these elements were bound by OTX2 (Supplementary Figure 8f). This suggests that DUX4 has a more dominant role than OTX4 in the activation of MLT2A1. This is particularly interesting because MLT2A1 is part of HERVL, and both have been reported to be upregulated upon DUX4 overexpression in ESCs[19]. We also consistently observed activation of HERVL transcription at the 4-cell stage (Fig. 4g). Collectively, our combined analyses of CA and GE datasets strongly suggest that DUX4 activates HERVL by binding to the MLT2A1 element, and that this is a dominant event in ERV activation during EGA (Fig. 4h).

## Discussion

Herein, we performed a genome-wide survey of accessible chromatin and GE in human pre-implantation embryos that revealed strong associations between accessible chromatin and the activation of EGA genes. These findings extend our knowledge of chromatin architecture and the sequential order of gene activation during early human embryogenesis. One challenge to this field has been to identify regulators that participate in the initiation of pre-implantation development. In the present study, by integrating the two omics data sets, we provided evidence showing that DUX4 induces remodeling of chromatin structure at both EGA genes and MLT2A1 elements during human pre-implantation development. However, we also suggest that DUX4 is not the earliest regulator, because the GE data show that it is likely not a maternally inherited transcript (Supplementary Figure 6c). Further investigations with additional complementary approaches are warranted to identify the factors that activate DUX4 expression. Another interesting finding at earlier stages was the transient opening of chromatin at the zygote stage. We have proposed the possibility that the presence of maternal TFs, rather than paternal genome accessibility, is likely the major driver of this transient change. Chromatin opening at the zygote

stage might than, be important for the access of key TFs, enabling widespread remodeling of chromatin structure during later developmental stages. The extensive remodeling of chromatin during embryonic development indicates that it could be very sensitive to disruption, which is associated with developmental abnormalities. Thus, our method may enable us to infer disruption of chromatin remodeling, providing the foundation for a promising pre-implantation genetic diagnostic test in the near future.

During the preparation and review of our manuscript, other groups also reported CA profiling of early human embryos[34–36]. Some observations, however, were not identical across our and their studies. For example, we and Wu et al.[35] found high enrichment of DUX4 at very early stages (2- to 4-cell stages), and of OCT4 at later stages (morula and ICM). In contrast, the study by Gao et al.[36] reported significant OCT4 enrichment in both the 8-cell and morula stages, and they did not show binding of DUX4 at an early stage. Disparate conclusions may be due to the use of different enzymes in the profiling of CA. For example, the hypersensitive form of Tn5 transposase might be able to capture the accessible regions more efficiently than DNase I endonuclease, especially when the input of embryonic material is extremely low and the chromatin within early stage embryos is largely inaccessible. Thus, our data may be helpful in providing insights into epigenetic events shortly after fertilization. In addition, the simultaneous profiling of both CA and the transcriptome in our study allowed us to integrate the two omics datasets; and we thus anticipate that this approach will in the future be extremely valuable regarding mechanistic studies of human pre-implantation development.

## Methods

**Informed consents.** This study was approved and guided by the Ethics Committee of the Reproductive & Genetic Hospital of CITIC-XIANGYA (Research license LL-SC-2016–015). All of the gametes and embryos were collected voluntarily after obtaining written informed consent from donor couples, who were undergoing in vitro fertilization (IVF) treatments at the Reproductive & Genetic Hospital of CITIC-XIANGYA, using standard clinical protocols as described previously[37]. Infertility in these donor couples was purely owing to tubal factors, and the women were under 35 years of age. Couples who donated eggs were informed that the donation posed a potential risk to their fertility success for that cycle. No financial benefit was involved in the donation process.

**hESC culture and differentiation.** hESCs were cultured as previous described[38]. In brief, hESCs were cultured on a feeder layer of mitotically inactivated mouse embryonic fibroblasts. The basic culture medium for maintenance of hESCs consisted of Dulbecco's modified Eagle's medium/F-12 (Gibco) supplemented with 15% knockout serum replacement (KSR, Invitrogen), 2 mM non-essential amino acids (NEAA, Invitrogen), 2 mM L-glutamine (Invitrogen), 0.1 mM β-mercaptoethanol (Invitrogen), and 4 ng/ml of basic fibroblast growth factor (Invitrogen). The medium was changed daily and hESCs were mechanically passaged every 6–7 days, and then differentiated toward hepatocyte-like cells as described previously[39]. In brief, hESCs were passaged onto Matrigel and cultured in mTeSR (STEMCELL Technologies) until a confluence of 50–70% was attained. Then, we cultured cells in RPMI-1640 (Gibco) supplemented with 25 ng/ml Wnt 3a (R&D Systems) and 100 ng/ml activin A (R&D Systems) for 3 days. To induce hepatic endoderm, cells were grown in KO/DMEM medium (Gibco) supplemented with 2% fetal bovine serum (Gibco) and 25 nM/ml KGF (R&D Systems) for 2 days, and then further cultured in SR/DMSO (KO/DMEM containing 20% KSR, 1% NEAA, 1% dimethylsulfoxide (Sigma), 1 mM L-glutamine, and 0.1 mM β-mercaptoethanol) for 4 days. The final maturation step involved culturing the cells in KO/DMEM medium that was supplemented with 10% FBS, 10 ng/ml Oncostatin M (OSM, R&D Systems), 10 ng/ml hepatocyte growth factor (R&D Systems), and 0.5 μM dexamethasone (R&D Systems) for 7 additional days. We then enzymatically digested the differentiated hepatocyte-like cells and undifferentiated hESCs to create a single-cell suspension. Ten or 50,000 cells were used for LiCAT-seq or ATAC-seq library preparation, respectively.

**Preparation of mouse pre-implantation embryos.** All of the animal studies were performed in accordance with the guidelines promulgated by the Institutional Animal Care and Use Committee of Central South University. C57BL/6 N female mice (6–8 weeks old) were superovulated by injection with 10 IU of pregnant mare

serum gonadotropin (PMSG, San-Sheng Pharmaceutical Co., Ltd.), followed by an injection with 10 IU of human chorionic gonadotropin (hCG, San-Sheng Pharmaceutical Co., Ltd.) 48 h later. We then mated the superovulated female mice with C57BL/6 N male mice. We subsequently collected 4-cell and morula-stage embryos from the oviducts of female mice 54–56 h or 76–78 h after hCG administration, respectively.

**Collection and culture of human oocytes and embryos**. Oocytes were donated by couples who had > 20 oocytes derived from the same IVF cycle. The cumulus cells surrounding the oocytes were removed by hyaluronidase (Vitrolife) treatment, and only mature MII oocytes were used in this study. We produced zygotes, and 2- and 4-cell-stage embryos by conventional intracytoplasmic sperm injection of donated oocytes by donated sperm from the same couple. Embryos were transferred to the wells of pre-equilibrated EmbryoSlides (Vitrolife) and cultured in G-1 Plus media (Vitrolife). Slides containing embryos were placed into the Embryoscope chamber immediately and cultured at 37.5 °C in 6% $CO_2$, 5% $O_2$, and 89% $N_2$ at high humidity. Day-3 cleavage-stage embryos were donated by couples who already had 2 healthy babies through IVF and wished to donate the surplus frozen embryos. Embryos were warmed according to the manufacturer's protocols (Kitazato Biopharma), and warmed embryos were transferred to the wells of a pre-equilibrated EmbryoSlide and cultured in G-2 Plus medium (Vitrolife). Slides containing embryos were placed into the Embryoscope chamber immediately and cultured at 37.5 °C in 6% $CO_2$, 5% $O_2$, and 89% $N_2$. We collected 8-cell, morula, and blastocyst-stage embryos at 2 h, 24 h, and 72 h after warming, respectively. All of the embryos used in the present study possessed good morphology and followed appropriate developmental timing. Embryonic assessment was performed as described previously[3].

**Sperm preparation**. We obtained human spermatozoal samples from the sperm bank of the Reproductive and Genetic Hospital of CITIC-Xiangya. After thawing, sperm were prepared by density gradient centrifugation using SpermGrad (Vitrolife), and those with vigorous activity were selected for preparation of our LiCAT-seq library.

**Isolation and separation of individual embryo cells**. Selected oocytes or embryos were transferred to acidic Tyrode's solution (pH 2.5, Sigma) to remove the zona pellucida. Zona-free embryos were then incubated for 10 min (for the 2-, 4-, and 8-cell stage embryos) or 20 min (for morulae) in Accutase medium (Gibco) before dissociation into single embryo cells by careful pipetting. We then manually separated blastocysts into ICM and TE using laser microdissection. ICM and TE compartments were subsequently placed into Accutase medium for 30 min and washed thoroughly in phosphate-buffered saline (PBS) with 0.5% (m/v) bovine serum albumin (BSA) (Sigma). We isolated single embryo cells by gentle, repeated pipetting; washed them 3–5 times in PBS with 0.5% BSA; and placed the cells into a PCR tube for LiCAT-seq library preparation.

**Bulk ATAC-seq**. We prepared bulk ATAC-seq libraries using a modified protocol based on a previous study[14]. In brief, 50,000 cells were collected, washed with cold 1× PBS and centrifuged and resuspended using 50 μl of ice-cold lysis buffer (10 mM Tris-HCl, pH 7.5; 10 mM NaCl, 3 mM $MgCl_2$ and 0.1% (v/v) IGEPAL CA-630 [Sigma]). The lysate was then centrifuged and resuspended in 50 μl of transposase reaction mix (10 μl 5× TAG buffer (50 mM TAPS-NaOH, pH 8.5; 25 mM $MgCl_2$, 50% [v/v] DMF), 1.5 μl of in-house Tn5 transposase and nuclease-free water (NF-water)), and the mix was incubated for 30 min at 37 °C. The subsequent steps of the protocol were performed as described previously[14].

**LiCAT-seq**. We isolated blastomeres or cells using a mouth pipette and transferred them into a 200-μl PCR tube, and lysed cells in 6.5 μl of mild lysis buffer (10 mM NaCl, 10 mM Tris-HCl, pH 7.5; 0.5% IGEPAL CA-630, 0.25 μl of 40 U/μl RNase-inhibitor [NEB] and NF-water) for 30 min at 4 °C. The lysate was vortexed for 1 min and then centrifuged at 2000 g for 5 min in a refrigerated centrifuge to force nuclei to the bottom of the tube. 4 μl of lysis product supernatant (containing the RNA) was carefully transferred into another tube supplemented with 0.5 μl of ERCC spike-in mixture (1:250,000 dilution in NF-water, Ambion), 1 μl of 10 mM dNTP mix and 1 μl of 10 μM modified oligo-dT primer (5′-AAGCAGTGGTATC AACGCAGAGTACT$_{30}$VN-3′, where V was either A, C, or G, and N was any base); and we then incubated at 72 °C for 3 min. Note that the physical separation procedure was critical for the successful capture of chromatin and RNA content.

Immediately after the separation step, 8.5 μl of a reverse-transcription master mix (0.75 μl of 200 U/μl SuperScript II reverse transcriptase (Invitrogen), 0.375 μl of 40 U/μl RNase-inhibitor, 1× Superscript II First-Strand buffer, 0.75 μl of 0.1 M dithiothreitol, 3 μl of 5 M betaine (Sigma), 0.45 μl of 0.2 M $MgCl_2$, 0.15 μl of 100 μM Template-Switching Oligo (5′-AAGCAGTGGTATCAACGCAGAGTACATrGrG+G-3′, where "r" indicates a ribonucleic acid base and " + " indicates a locked nucleic acid base, Exiqon) and NF-water) were added to the tube. The mixture was then thermocycled as follows: 42 °C for 90 min, 10 cycles of 50 °C for 2 min, 42 °C for 2 min, and finally 70 °C for 15 min. Afterward, the PCR master mix (15 μl KAPA HiFi HotStart ReadyMix with 0.3 μl of 10 μM PCR primer [5′-AAGCA GTGGTATCAACGCAGAGT-3′]) was added to the 15 μl of reverse-transcription

reaction mixture and thermocycled as follows: 98 °C for 3 min, 18 cycles of 98 °C for 20 s, 67 °C for 20 s, 72 °C for 6 min, and finally 72 °C for 5 min. Amplified cDNA was purified using a 1:1 volumetric ratio of AMPure XP beads (Beckman Coulter) and then eluted into 20 μl of NF-water.

During the RNA library preparation process, the precipitated nuclei were resuspended in a 4 μl transposase reaction mix (1.4 μl 5× TAG buffer, 0.7 μl Tn5 transposase [in-house] and NF-water). The transposition reaction was carried out for 30 min at 37 °C. Then, we added 3.5 μl mix of stop buffer (2.1 μl of 0.1 M ethylenediaminetetraacetic acid, pH 8.0, 0.42 μl of 0.1 M Tris-HCl, pH 8.0 and NF-water) and the reaction was maintained at 50 °C for 30 min. The nuclei were lysed by adding strong lysis buffer (6 μl of RLT Plus buffer [QIAGEN]) to the mixture after the stop step, with the lysis process performed by shaking on a thermomixer for 15 min at 37 °C. We then purified the DNA using a 1:1.8 volumetric ratio of AMPure XP beads, and finally eluted the DNA with 10 μl of NF-water. We used a 20 μl PCR amplification master mix (9 μl of transposed DNA, 10 μl of NEBNext High-Fidelity 2× PCR Master Mix, 0.5 μl of 20 μM transposase adapter 1 (5′-TCGTCGGCAGCGTCAGATGTGTATAAGAGACAG-3′), and 0.5 μl of 20 μM adapter 2 (5′-GTCTCGTGGGCTCGGAGATGTGTATAAGAGACAG-3′]) to amplify the DNA and then proceeded to perform 8 cycles of PCR using the following conditions: 72 °C for 5 min; 98 °C for 1 min; and thermocycling at 98 °C for 15 s, 63 °C for 30 s, and 72 °C for 1 min. The pre-amplified transposed DNA was harvested using a 1:1 volumetric ratio of AMPure XP beads and finally eluted in a total of 25 μl of NF-water.

For CA libraries, we amplified the DNA for another 8–10 cycles (the numbers of cycle were designated by qPCR analysis for different cell types according to a previous study[14]) using the following PCR reaction mixture: 23 μl of pre-amplified transposed DNA, 25 μl of NEBNext High-Fidelity 2× PCR Master Mix, 1 μl of 20 μM universal primer, and 1 μl of 20 μM barcoded primer. For sequencing, size-selected with AMPure XP beads for fragments between 150 and 350 bp in length according to the manufacturer's instruction, and finally eluted with 20 μl of TE buffer. For RNA libraries, 2 ng of cDNA were used for the tagmentation reaction carried out with a 10 μl mixture containing 0.3 μl of Tn5 transposase, 1× TAG buffer, and NF-water. The tagmentation reaction was incubated at 55 °C for 10 min and we released Tn5 with 2.5 μl of 0.1% sodium dodecyl sulphate. The transposed cDNA was then used for PCR amplification and library preparation according to the Smart-seq2 method described previously[13].

**Library preparation and sequencing**. All of the libraries were further prepared based on a BGISEQ-500 sequencing platform[40]. In brief, the DNA concentration was determined by Qubit (Invitrogen), and 2 pmol of pooled samples were used to make single-strand DNA circles (ssDNA circles). We then generated DNA nanoballs (DNBs) with the ssDNA circles by rolling circle replication to increase the fluorescent signals at the sequencing process as previously described[40]. The DNBs were loaded onto the patterned nanoarrays and sequenced on the BGISEQ-500 sequencing platform with single-end 50-bp read lengths.

**Transcriptome data processing**. Raw reads were first aligned to the human rRNA sequence, including 28 S (NR_003287.2), 18 S (NR_003286.2), 5 S (NR_023379.1), and 5.8 S (NR_003285.2), using SOAP2[41] and the mapped reads were filtered using custom scripts. The retained reads were mapped to the hg19 genome using Hisat2[42] with the following parameters: --sensitive --no-discordant --no-mixed -I1 -X1000, and reads with mapping quality < 30, or duplicate reads were discarded using samtools[43]. Reads within genes (GENCODE, v19) were counted using the GenomicAlignments package[44] with the following parameters: mode = "Union", inter.feature = TRUE and singleEnd = TRUE. The raw read counts were then normalized by "rlogTransformation" function of DESeq2[45].

**CA data processing**. The CA data were trimmed by custom scripts and aligned using Bowtie[46] (parameter: -X2000 -m1). Reads with mapping quality < 30, and reads mapped to the mitochondria genome or the hg19 consensus excludable region (http://hgdownload.cse.ucsc.edu/goldenPath/hg19/encodeDCC/wgEncodeMapability/) were filtered. We removed duplicate reads using Mark-Duplicates function of Picard (http://broadinstitute.github.io/picard/). For peak calling, we used the model-based analysis of ChIP-seq (MACS2)[47] to identify the read enrichment regions (peaks) of the CA with the following parameters: -g hs -B -q 0.01 --nomodel --nolambda --extsize 250. Only peaks detected in both replicates were used for downstream analysis, and peaks located on sex chromosomes were excluded. Peaks for all of the stages were merged together to a unique peak list, and the number of raw reads mapped to each peak was quantified using bedtools[48]. We then further normalized the raw read counts with the "rlog-Transformation" function of DESeq2[45]. We also used deepTools2[49] to compute and visualize the TSS enrichment for each CA profile; and for signal tracks, the p values for MACS2-called narrow peaks were used to filter out the noise reads.

**Calculation of promoter GC content**. We calculated a promoter GC ratio for regions ± 500 bp from the TSS. The promoters were then separated into HCPs, ICPs, and LCPs based upon the GC content cutoffs as follows: HCP:GC ratio ≧ 0.75; ICP:GC ratio < 0.75, and ≧ 0.25; LCP: GC ratio < 0.25.

**Fuzzy clustering analysis.** We used the mean value of the DEseq2-normalized CA level for the biological replicates as an input for Fuzzy analysis[18]. Prior to clustering, we removed peaks with normalized values < 0 at all stages. Then, time-course c-means Fuzzy clustering was performed with 6 centers and a cluster membership threshold of 0.3. A gene was assigned to a peak if the peak was located 10 kb up- or downstream of the gene. EGA genes were defined as genes that expressed (DEseq2-normalized value > 0) in at least one of the 2-cell to morula stages, and showed fold-changes greater than twice that of oocytes.

**Motif-enrichment analysis.** For Fig. 3a, we performed motif-enrichment analysis for gained CA peak regions using HOMER2[23] for known TF motifs with the option: -size 200. TFs with both high motif-enrichment ($-\log p$ value > 100) and GE level (DESeq2-normalized expression value > 0) in at least 1 stage are shown. For further validation analyses of DUX4 enrichment, the published DUX4 ChIP-seq data in hESCs were downloaded (GSE94322);[28] and bedtools (intersect) was used to count the intersection of DUX4-binding sites and accessible regions at each stage.

**Accessibility analysis of retrotransposon elements.** We downloaded coordinates and annotations of retrotransposon elements from the UCSC Genome Browser (open-4-0-7 version of RepeatMasker[50]). The number of raw reads mapped to each element was then counted using bedtools[48]. We normalized the read counts by the length of the retrotransposon element (after merging in kilobases) and by the sum of the reads mapping to all retrotransposon elements (in millions). Normalized read counts were then log-transformed after adding a pseudo-count of 1. We used a Shannon-entropy-based method to identify stage-specific retrotransposons as previously described[51]. We treated retrotransposons with top entropy scores as stage-specific retrotransposons. For motif-enrichment analyses of the candidate retrotransposon elements, HOMER2[23] was used for de novo TF motif prediction with the option: size 200.

**Code availability.** The R scripts used in this study are available at https://github.com/single-cell-BGI/Human_Embryo_LiCAT.

**Reporting Summary.** Further information on experimental design is available in the Nature Research Reporting Summary linked to this article.

## Data availability

All of the raw data have been deposited to CNGB Nucleotide Sequence Archive (accession code: CNP0000193; https://db.cngb.org/cnsa/project/CNP0000193/public/). All of the raw data have also been deposited to NCBI Sequence Read Archive (accession code: SRP163205; https://www.ncbi.nlm.nih.gov/sra/?term = SRP163205). The source data for generating the figures are provided as Supplementary Data files 1–4. All other relevant data supporting the key findings of this study are available within the article and its Supplementary Information files or from the corresponding author upon reasonable request. A reporting summary for this Article is available as a Supplementary Information file.

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

## Acknowledgements

We thank the patients and staff at the Reproductive & Genetic Hospital of CITIC-Xiangya for their assistance. We thank all members of the Stem Cell and Development Lab (BGI) for their support. We thank Dr. J. Lynn Fink for helpful comments. This work was supported by Shenzhen Peacock Plan (KQTD20150330171505310), the National Natural Science Foundation of China (No. 81471510), the Shenzhen Municipal Government of China (GJHS20170314152802146), and the National Key Research and Development Program of China (Project No. 2016YFC1000200).

## Author contributions

L.L., L. Leng, C.L., Y.Y., Z.S., and G.L. conceived the idea and designed the entire project. L. Leng, C.Lu, and L.Z. contributed to oocyte collection, sperm treatment, and embryo culture in vitro. F.G., L.H., and G.L. performed clinical conversation and the signing of the informed consents. C.L. and Y.Y. developed and optimized the LiCAT-seq method. C.L. and Y.Y. generated the CA and GE data. L.L. performed the bioinformatic analysis with the assistance of C.L., Y.Y., and L.W. F.G., S.Z., X.W., M.W., L.Z., and L.H. assisted with the experiments. J.W., H.Y., S.Z., F.C., and G.L. advised the development of LiCAT-seq technology and data analysis. G.L. and Z.S. supervised the entire study. L.L. and L. Leng wrote the manuscript with input from C.L. and Y.Y. All authors read and approved the manuscript for submission.

## Additional information

**Competing interests:** The authors declare no competing interests.

