## [Peer Review File · Nature Communications]

Reviewer #1 (Remarks to the Author):

This is a well conducted albeit essentially descriptive study, which provides information on the early steps of human embryogenesis. Its strong points are the apparent efficiency of the technique promoted here, the soundness of most of the bioinformatics analyses, and the description of transiently chromatin accessible sites prior to and during EGA. Its weak points are the lack of functional data and the largely confirmatory nature of the results, after the publication last year of a series of Nature Genetics papers unveiling the role of Dux4 in EGA and more recently of a Gao et al. Cell 2018 study describing not only results fairly similar to those reported here (albeit with less insights about zygotes) but also those of DNA methylation analyses, functional experiments and evolutionary comparisons.

In addition, I think the following specific points should be addressed:

Major:

1. The total number of reads mapped without mitochondrial DNA and PCR duplicates should be presented in a supplementary table to see how the latter impact on the results and their interpretation.
2. Authors combined replicates, which prevented them from providing statistics. This should be reverted, and statistical significance of all results should be formally assessed.
3. Can the authors exclude that results related to CpG content reflect a Tn5-, PCR- or sequencing-derived artefact?
4. The transient opening of chromatin at the zygote stage (Fig. 1b) is the most interesting aspect of the paper, because it has not been reported before. It should be investigated further. Could it be due to the maternal presence of transcription factors or to paternal genome accessibility?
5. The preference for accessibility at promoters up to 2-cells and then at distal regions from 4 to 8-cells is also intriguing (Fig1d, e). Is this due to the fact that all promoters are opened at 2-cell stage which inevitably renders additional opened regions distal ?
6. The statement that there are more changes in chromatin accessibility than expression between zygote and 4-cell stage is definitively not obvious from the PCA illustrated in Fig1f.
7. Fig. 2 and its conclusions do not make sense, at least as presented. Chromatin accessibility is relevant only at TSS. Furthermore, the datasets should be handled the other way around, clustering the EGA-genes and then examining their chromatin accessibility pattern. As done, the figure does not demonstrate any clear correlation between chromatin accessibility and gene expression except maybe for groups C1 and C3 (and only for EGA genes therein). The statement "Our analyses reveal a strong relationship between chromatin accessibility and EGA-only genes activation" is thus not supported. Moreover, all clusters seem to have fairly similar patterns of gene expression, as if they had been randomly selected. Supporting this criticism, how can there be an even distribution of EGA genes amongst clusters that display distinct patterns of chromatin accessibility? If anything it indicates a lack of correlation between the two parameters.

Minor:

1. How were the heat maps (Fig1b, S1a, S4b) and genome browser views (F1b, F3e, S1b, S2f) normalized? There is a higher background for the oocyte track: is this due to uneven scaling?
2. Would normalization for number of reads and number of haploid genomes (possible since the authors know the number of used blastomeres) yield the same results (Supp. table 1)?
3. "Supporting this, genes in these two clusters include regulators such as KDM4E, KLF17 and ZSCAN4, which have been reported to be the early regulators of mouse preimplantation development 15." The authors reference a paper which never showed the role of those genes in early embryonic development.
4. "LTR-lacking elements (such as LINEs, SINES and SVA elements)": SINES, not SINES.
5. "levels of LTR5_Hs and LTR7B and the corresponding HERVK and HERVH elements, which have been reported to be crucial during early embryo development and establishment of naïve pluripotency^{29, 30}". Neither HERVK nor HERVH has been formally demonstrated to be essential for early embryogenesis. The only data available so far are for a role of HERVH in the maintenance of primed human embryonic stem cell pluripotency.
6. Fig.3: How were DUX4 target genes selected?
7. Fig.4 is purely descriptive and reports findings for most already published in recent Dux4-related papers.

Reviewer #2 (Remarks to the Author):

This manuscript presents data on the development of a technique (LiCAP-seq) to simultaneously analyze both chromatin accessibility and the transcriptome using a very small number of human embryo cells. The technique appears to work well as high quality of data are obtained and presented. The authors subsequently investigate human preimplantation embryo using this new platform.

Better understanding of human preimplantation embryos has important implications in fertility and the relevant disease. Due to the ethical and technical challenges, analyzing the chromatin structure in preimplantation embryos was not possible until recently.

The current study has collected a huge amount of data on genome-wide chromatin accessibility and transcriptome of several stages of human preimplantation embryos, oocytes, the ICM and the TE. The information will be very valuable for dissecting the molecular mechanism of human development in general and for exploring establishment of better human pluripotent stem cells in vitro in particular.

Specific comments:

1. The manuscript has systematically investigated chromatin accessibility and the transcriptome of the human preimplantation embryo cells, it will substantially improve the manuscript if more analysis results of the datasets can be presented. For example, besides those open chromatin accessible regions gained in development as shown in the manuscript, what about those lost ones?
2. The significance of some findings in the study is not emphasized. Example: in Figure 4a, the

chromatin accessibility at the retrotransposon regions demarcates the embryo stages from 4-cell to the blastocyst clearly, in contrast to retrotransposon expression.

3. The study confirms that DUX4 is an important regulator of ZGA in human as its motifs are enriched in open chromatin regions of many of its target genes. Have the authors analyzed the DUX4 locus for open chromatin accessibility regions, and the enriched motifs for transcription factors that may regulate DUX4 expression prior to ZGA?

4. Figure 3 on enrichment of TF motifs within the gained accessible regions, is the list of TFs shown the complete one? It will be more meaningful if those TFs known to be important in mouse preimplantation embryos are investigated. For example, Hippo/Tead/Yap1, Nr5a2 and Rarg are important in the lineage segregation of the ICM and the TE in the mouse. Are their binding motifs enriched in any accessible regions in human preimplantation embryos? If not, it may indicate species difference in this important development process.

5. Supplementary figure 3b shows chromatin accessible regions near those genes encoding epigenetic modifier genes that are known to have key roles in mouse preimplantation embryos. What's the significance of these results in the context of human preimplantation embryo development?

6. The recent Cell paper (Gao et al 2018) on DHS landscape of human preimplantation embryo identified that OCT4 contributed to zygotic genome activation in humans. Is this conclusion supported by the current study?

Reviewer #3 (Remarks to the Author):

This manuscript is entitled "Simultaneous profiling of chromatin accessibility and the transcriptome of human preimplantation embryos reveals widespread regulatory rewiring during embryonic genome activation." The manuscript reports the development of a new technique called LiCAT-seq (low input chromatin accessibility and transcriptome sequencing) to potentially enable profiling of developmental processes, in human embryos, that were previous unexplored due to shortage of biological material through development. The manuscript has several limitations:

1) There is a need to edit carefully. The abbreviations for LiCAT-seq, DUX4 and others should be explained the first time they are used (in the abstract). The references for the manuscript, as well, to include more references to human embryogenesis (since the paper is on human preimplantation embryo development). Finally, the title of the manuscript includes the phrase "reveals widespread regulatory rewiring" and it is not clear what that means nor has the manuscript demonstrated regulatory rewiring (the manuscript uses correlation analysis).

2) A new assay is developed here for use in human embryo developmental studies. There is minimal validation of the assay. It should be used in human embryonic stem cells, other mammalian embryos (mouse for example) or comparable systems. The assay should have positive and negative controls that might include genetically-modified (null) mutations that can be used to demonstrate reliability in small cell numbers (reflecting larger cell populations).

3) The parameters of LiCAT-seq that enable use of the assay with small cell numbers (relative to traditional assays) should be described in more detail. What enabled use of this assay when others have failed?

4) The number of oocytes and embryos used must be included, along with the history (from the same woman, couple, or not; general morphology or indicators of viability; reason for donation (family has already been established, supernumerary, etc).

5) The table describing blastomere number (supplementary table 1) does not make much sense. One cannot collect 10 blastomeres from an oocyte, for example. The table should be clarified.

On page 10 of the manuscript, the essence of the work is described by the authors: "In summary, we performed a genome-wide survey of accessible chromatin....." "We revealed strong associations...." This is a research paper that describes a survey method that has uncovered associations. Overall, the data is not proving the title of the manuscript but suggests the implications of the title. If the issues above were addressed, it might be more suitable for publication.

Reviewers' comments:

Reviewer #1 (Remarks to the Author):

This is a well conducted albeit essentially descriptive study, which provides information on the early steps of human embryogenesis. Its strong points are the apparent efficiency of the technique promoted here, the soundness of most of the bioinformatics analyses, and the description of transiently chromatin accessible sites prior to and during EGA. Its weak points are the lack of functional data and the largely confirmatory nature of the results, after the publication last year of a series of Nature Genetics papers unveiling the role of Dux4 in EGA and more recently of a Gao et al. Cell 2018 study describing not only results fairly similar to those reported here (albeit with less insights about zygotes) but also those of DNA methylation analyses, functional experiments and evolutionary comparisons.

In addition, I think the following specific points should be addressed:

Major:

1. The total number of reads mapped without mitochondrial DNA and PCR duplicates should be presented in a supplementary table to see how the latter impact on the results and their interpretation.

Answer: We have now added a supplementary table (Supplementary Table 2) showing the summary statistics of the total number of usable reads for the embryo datasets. To further validate our low-input method, we have applied our method to profile other cell types including human pluripotent/differentiated cells and mouse embryos (Supplementary Fig. 1 and 2). The summary statistics of sequencing datasets is also included in Supplementary Table 2.

2. Authors combined replicates, which prevented them from providing statistics. This should be reverted, and statistical significance of all results should be formally assessed.

Answer: We have now replaced many of the graphs by using data from both replicates (e.g., Fig. 1e, Fig. 2c, d, Supplementary Fig. 1, Supplementary Fig. 2, Supplementary Fig. 5d-f, Supplementary Fig. 6c, f). For visualization of example regions, we also used both replicates in the revised manuscript (e.g., Fig. 1d, Supplementary Fig. 1h, Supplementary Fig. 2d).

3. Can the authors exclude that results related to CpG content reflect a Tn5-, PCR- or sequencing-derived artefact?

Answer: A previous study of mouse embryos indicated that the regions with a high proportion of GC content tend to be accessible¹ (see reference below), which is consistent with our data from human embryos (Supplementary Fig. 3e). To further assess whether this is a technical artefact, we have now compared the chromatin accessibility signal on regions with similar level of GC content in human embryonic stem cells. We found that even on the regions of same level of GC content, the known open regions (DNase I hypersensitive sites) still show a dramatically higher accessibility level (Supplementary Fig. 1c). This indicated that our technique can faithfully detect the regions with high accessibility.

4. The transient opening of chromatin at the zygote stage (Fig. 1b) is the most interesting aspect of the paper, because it has not been reported before. It should be investigated further. Could it be due to the maternal presence of transcription factors or to paternal genome accessibility?

Answer: This is a very interesting question. We agree that the presence of maternal transcription factors or paternal genome accessibility may be the reasons for the transient opening of zygote chromatin. To validate these hypotheses, we firstly profiled the chromatin accessibility for mature sperm. Surprisingly, while we observed high accessibility in sperm chromatin, it presents the opposite pattern with all other stages (including zygote). For example, the reads in sperm profiles are extensively enriched in gene deserts, whereas the reads in the profiles of the other stages are enriched in gene-rich regions (Fig. 1d, e). This observation is supported by an earlier study showing preferential nucleosome retention at gene deserts in sperm relative to other cell types² (see reference below). Moreover, we found that sperm chromatin presents much lower accessibility on zygote opening regions than oocyte chromatin (Supplementary Fig. 4c). These results indicated that the paternal genome has been undergone complete reprogramming during fertilization and is not likely contributing to the transient opening of zygote chromatin. To assess whether maternal TFs could be the reason, we next predicted the candidate TFs that potentially bind to the open regions in zygote genome. We found that most of the top enriched TFs are also highly expressed in zygote, such as SP1, KLF5, NFYA (Supplementary Fig. 4d, e). Therefore, the maternal presence of TFs could be one explanation of transient opening of zygote genome. The sharp decrease of accessibility at 2-cell stage is also intriguing. It may due to the degradation of TFs or co-factors of maternal origin during the maternal-to-zygote transition process, which would be interesting to investigate with more elaborate time points in further studies. We have now added descriptions on this interesting question in the main text (page 5 lines 5-29).

5. The preference for accessibility at promoters up to 2-cells and then at distal regions from 4 to 8-cells is also intriguing (Fig1d, e). Is this due to the fact that all promoters are opened at 2-cell stage which inevitably renders additional opened regions distal?

Answer: This is an interesting point. In the revised manuscript, a peak in a certain stage was selected only when it is present in both replicates, which is more strict than our previous approach (where peaks were called from reads merged from replicates). Using this method, we found that both the sperm and oocyte showed strong enrichment of peaks in distal, rather than promoter proximal regions, which is reasonable because genes are protected in hyper-methylated regions³ (see reference below). Regarding this question, although the proportion of promoter peaks decreases (from ~50% at the 2-cell stage to ~40% at the morula stage), the exact number of peaks still increases (the number of peaks in morula stage is 10-fold higher than that in the 2-cell stage). However, when analyzing the dynamics of promoter and distal peaks of EGA genes, we found that the promoter peaks increase more strikingly than distal peaks at the 4-cell stage, but both types of peaks present similar accessibility levels at other stages, suggesting that promoters opened at the 4-cell stage may render opened distal regions at later stages. We have now added this explanation in the main text (page 7, lines 8-12).

6. The statement that there are more changes in chromatin accessibility than expression between zygote and 4-cell stage is definitively not obvious from the PCA illustrated in Fig1f.

Answer: We agree with this reviewer and have now changed the description in the main text (page 6, lines 1-6).

7. Fig. 2 and its conclusions do not make sense, at least as presented. Chromatin accessibility is relevant only at TSS. Furthermore, the datasets should be handled the other way around, clustering the EGA-genes and then examining their chromatin accessibility pattern. As done, the figure does not demonstrate any clear correlation between chromatin accessibility and gene expression except maybe for groups C1 and C3 (and only for EGA genes therein). The statement “Our analyses reveal a strong relationship between chromatin accessibility and EGA-only genes activation” is thus not supported. Moreover, all clusters seem to have fairly similar patterns of gene expression, as if they had been randomly selected. Supporting this criticism, how can there be an even distribution of EGA genes amongst clusters that display distinct patterns of chromatin accessibility? If anything it indicates a lack of correlation between the two parameters.

Answer: We thank the reviewer for these helpful suggestions that improved our manuscript. As suggested, we have now changed the analysis strategy by clustering EGA genes, followed by examination of accessibility dynamics of both promoter proximal and distal peaks. Interestingly, the EGA genes could be classified into six clusters based on the expression dynamics. The overall chromatin accessibility changes are quite similar with the expression changes in five of the clusters (C1-5), revealing the essential role of chromatin reconfiguration in the activation of EGA-only genes. Notably, the changes in promoter proximal peaks are more striking than distal peaks at the 4-cell stage, indicating distinct contributions of promoter and enhancer regions

during EGA. We have now changed the figures (Fig. 2), as well as the description (page 7, lines 1-30 and page 8, lines 1-6) in the revised manuscript.

Minor:

1. How were the heat maps (Fig1b, S1a, S4b) and genome browser views (F1b, F3e, S1b, S2f) normalized? There is a higher background for the oocyte track: is this due to uneven scaling?

Answer: The heat maps and genome browser viewers are normalized by RPKM. The higher background for the oocyte track is likely due to the extensively inaccessible features of oocyte chromatin (only 54 detectable accessible regions), resulting in amplification of background noise during sequencing. Even so, the significant enrichment of reads in the TSS regions suggests successful capture of accessible regions. To avoid possible confusion, we have now applied a new strategy to perform the normalization for the genome browser views, which applied the p-value of each MACS2⁴ (see reference below) called peak to filter out the background noise (Fig. 1d, Fig. 3e, Fig. 4h and Supplementary Fig. 4a). This strategy has now been described in the Supplementary Methods.

2. Would normalization for number of reads and number of haploid genomes (possible since the authors know the number of used blastomeres) yield the same results (Supp. table 1)?

Answer: This is a quite interesting question. While the numbers of haploid genomes are quite similar across stages, the usable reads are not exactly similar to each other. This is likely due to different degrees of accessibility in each stage. For example, the same number of blastomeres was used at the oocyte and 4-cell stages, but the number of usable chromatin accessibility reads in the 4-cell stage is much higher than that in the oocyte. Despite this, it would be still interesting to see the results when different numbers of haploid genomes were used in the same stage/cell types. Because of limitations in obtaining human embryos, we performed the evaluation in a widely-used cell line (K562). Interestingly, we observed a linear relationship between normalization for the number of reads and the number of haploid genomes when the number of haploid genomes is less than 50 (see figure below), which indicated that these two normalization approaches could yield similar results. However, if a large number of cells was used, the number of usable reads would be less than expected, possibly due to incomplete capture of accessible regions in all haploid genomes.

3. "Supporting this, genes in these two clusters include regulators such as KDM4E, KLF17 and ZSCAN4, which have been reported to be the early regulators of mouse preimplantation development 15." The authors reference a paper which never showed the role of those genes in early embryonic development.

Answer: We agree with the reviewer and have now modified the statement to “Supporting this, genes in this group include regulators such as ZSCAN4, KDM4E and KLF17, which have been reported to be transcriptionally activated during human or mouse pre-implantation development” (page 7, lines 21-24).

4. "LTR-lacking elements (such as LINEs, SINES and SVA elements)": SINES, not SINES.

Answer: We have now modified this in the main text (page 11, line 5).

5. "levels of LTR5_Hs and LTR7B and the corresponding HERVK and HERVH elements, which have been reported to be crucial during early embryo development and establishment of naïve pluripotency^{29, 30}". Neither HERVK nor HERVH has been formally demonstrated to be essential for early embryogenesis. The only data available so far are for a role of HERVH in the maintenance of primed human embryonic stem cell pluripotency.

Answer: We agree with reviewer that these points were not clearly stated in previous manuscript. Early studies have shown that the essential role of primate-specific endogenous retrovirus HERVH in the establishment of naïve pluripotency⁵ (see reference below), and the role HERVK in the induction of viral restriction pathways in pluripotent stem cells⁶ (see reference below). Since HERVK is highly expressed at the 8-cell stage, the authors speculated that it might play similar roles during embryogenesis. We have modified the statement to “levels of LTR5_Hs and LTR7B and the corresponding HERVK and HERVH elements, which have been reported to be essential in the induction of viral restriction pathways and establishment of naïve pluripotency”

in the main text (page 11, lines 20-23).

6. Fig.3: How were DUX4 target genes selected?

Answer: The target genes of DUX4 were selected if the genes are located 10kb upstream of downstream of the DUX4 ChIP-Seq peaks. We have now explained this in the main text (page 10, lines 11-12).

7. Fig.4 is purely descriptive and reports findings for most already published in recent Dux4-related papers.

Answer: We agree with the reviewer that many of similar conclusions have been reported. However, all of the previous studies are mainly based on RNA-seq data. In our work, we showed that many of the key ERVs show synergistic dynamics on both chromatin accessibility and gene expression, suggesting the important role of chromatin remodeling, but not post-transcriptional regulation, in the activation of ERVs. We have now added more description in the main text to highlight this point (e.g., page 11, lines 8-9, lines 26-28 and page 13, lines 2-4).

Reviewer #2 (Remarks to the Author):

This manuscript presents data on the development of a technique (LiCAP-seq) to simultaneously analyze both chromatin accessibility and the transcriptome using a very small number of human embryo cells. The technique appears to work well as high quality of data are obtained and presented. The authors subsequently investigate human preimplantation embryo using this new platform.

Better understanding of human preimplantation embryos has important implications in fertility and the relevant disease. Due to the ethical and technical challenges, analyzing the chromatin structure in preimplantation embryos was not possible until recently.

The current study has collected a huge amount of data on genome-wide chromatin accessibility and transcriptome of several stages of human preimplantation embryos, oocytes, the ICM and the TE. The information will be very valuable for dissecting the molecular mechanism of human development in general and for exploring establishment of better human pluripotent stem cells in vitro in particular.

Specific comments:

1. The manuscript has systematically investigated chromatin accessibility and the transcriptome of the human preimplantation embryo cells, it will substantially improve the manuscript if more analysis results of the datasets can be presented. For example, besides those open chromatin accessible regions gained in development as shown in the manuscript, what about those lost ones?

Answer: We have now added more descriptive results on the embryo datasets. For example, the numbers of both gained and lost peaks in each stage were shown in Supplementary Fig. 6a. In addition, we included sperm accessibility datasets in the revised manuscript. Comparison of the read distribution between sperm and all other stages revealed a unique chromatin accessibility pattern in sperm (Fig. 1d, e). Moreover, to find the possible TFs that contribute to the transient opening of zygote genome, we investigated the enrichment of TF motifs, as well as the expression level of the TFs with top enrichment values, resulting in identification of the potentially important TFs at the zygote stage (Supplementary Fig. 4d, e).

2. The significance of some findings in the study is not emphasized. Example: in Figure 4a, the chromatin accessibility at the retrotransposon regions demarcates the embryo stages from 4-cell to the blastocyst clearly, in contrast to retrotransposon expression.

Answer: This is an interesting point that we did not demonstrate clearly in the previous manuscript. Briefly, this finding is due to the asynchronous dynamics between accessibility and expression for some of the ERVs. Although most of the ERVs are significantly enriched in specific stages (4-cell to TE of the blastocyst stage), some of the ERVs still show asynchronous dynamics on accessibility and expression, suggesting other factors (such as DNA methylation) may participate in the regulation of ERV elements. We have now added more description about this finding in the main text (page 11, lines 28-30 and page 12, lines 1-6). Another interesting finding is the transient opening of zygote chromatin. We have now explored this in the revised manuscript as well, which could be referred to Answer to Question 1 of the same reviewer, and Answer to Question 4 of Reviewer 1.

3. The study confirms that DUX4 is an important regulator of ZGA in human as its motifs are enriched in open chromatin regions of many of its target genes. Have the authors analyzed the DUX4 locus for open chromatin accessibility regions, and the enriched motifs for transcription factors that may regulate DUX4 expression prior to ZGA?

Answer: This is a very interesting and important question. Unfortunately, the challenge to address this question is that the *DUX4* gene is located in the telomere region of the chromosome 4 (4q35.2), which consists of series of repetitive DNA sequences. It is hard to accurately map the

NGS-based short reads to this region. Thus, measuring the accessibility of *DUX4* is impractical. We believe it would be interesting to explore the TFs that regulate *DUX4* using a long-reads based method in the future.

4. Figure 3 on enrichment of TF motifs within the gained accessible regions, is the list of TFs shown the complete one? It will be more meaningful if those TFs known to be important in mouse preimplantation embryos are investigated. For example, Hippo/Tead/Yap1, Nr5a2 and Rarg are important in the lineage segregation of the ICM and the TE in the mouse. Are their binding motifs enriched in any accessible regions in human preimplantation embryos? If not, it may indicate species difference in this important development process.

Answer: Because the complete list contains too many TFs motifs (more than 200) and many of them come from the same family, we chose the representative ones in the main figure. Considering the list of enriched TFs would be helpful for further study, we have now added a supplementary table containing all significantly enriched TF motifs in each stage. As expected, many of the TFs are reported in mouse embryo development (such as NANOG, POU5F1, DUX and TEAD). Interestingly, some of the TFs known to be important in mouse development, such as Nr5a2 and Rarg, are not enriched in human, suggesting species-specific regulatory mechanism. We have now added descriptions in the main text (page 9, lines 11-14).

5. Supplementary figure 3b shows chromatin accessible regions near those genes encoding epigenetic modifier genes that are known to have key roles in mouse preimplantation embryos. What's the significance of these results in the context of human preimplantation embryo development?

Answer: This is an interesting point. It is worth noting that a previous study on DNA methylome dynamics of human pre-implantation embryos showed dramatic *de novo* DNA methylation during the 4-cell to the 8-cell stage³. While investigating the CA and GE dynamics of DNA methyltransferases (DNMT1, DNMT3A, DNMT3B, DNMT3L) and demethylases (TET1, TET2, TET3), we observed quite similar dynamics on these two omics layers (Supplementary Fig. 5f). Interestingly, the *de novo* DNA methyltransferase DNMT3L showed a dramatic increase in both CA and GE, while the other enzymes from the same family showed the opposite trends, suggesting a potential role of DNMT3L in the regulation of *de novo* DNA methylation during EGA. Notably, the DNA demethylases also showed rather different patterns, for example, TET3 exhibits reduced expression though the entire process, which is supported by an earlier study showing the specific role of TET3 DNA dioxygenase in epigenetic reprogramming by oocytes⁷. In contrast, TET1 and TET2 are up-regulated during EGA, suggesting that the transition from DNA methylation to 5-hydroxymethylcytosine may occur during this process. We have now added these descriptions in the main text (page 8, lines 8-23).

6. The recent Cell paper (Gao et al 2018) on DHS landscape of human preimplantation embryo identified that OCT4 contributed to zygotic genome activation in humans. Is this

conclusion supported by the current study?

Answer: The Cell paper by Gao *et al.*⁸ (see reference below) showed dramatic increase of OCT4 enrichment at the 8-cell stage. However, in our study, OCT4 is mainly enriched in the morula and ICM stages. Although a contradictory result was observed, our results are supported by the conclusion in the Nature paper by Wu *et al.*⁹ (see reference below), which showed significant enrichment of OCT4 motif in ICM, but not the 8-cell stage⁹ (See reference below). Given that the data in our study and the study of Wu *et al.* are produced by a Tn5-based strategy, while Gao *et al.* used a DNase I-based strategy, the inconsistent conclusion is likely due to the difference between approaches. We have now discussed this in the discussion part of the main text (page 14, lines 4-16).

Reviewer #3 (Remarks to the Author):

This manuscript is entitled “Simultaneous profiling of chromatin accessibility and the transcriptome of human preimplantation embryos reveals widespread regulatory rewiring during embryonic genome activation.” The manuscript reports the development of a new technique called LiCAT-seq (low input chromatin accessibility and transcriptome sequencing) to potentially enable profiling of developmental processes, in human embryos, that were previous unexplored due to shortage of biological material through development. The manuscript has several limitations:

1) There is a need to edit carefully. The abbreviations for LiCAT-seq, DUX4 and others should be explained the first time they are used (in the abstract). The references for the manuscript, as well, to include more references to human embryogenesis (since the paper is on human preimplantation embryo development). Finally, the title of the manuscript includes the phrase “reveals widespread regulatory rewiring” and it is not clear what that means nor has the manuscript demonstrated regulatory rewiring (the manuscript uses correlation analysis).

Answer: We agree with the reviewer and have now modified these points accordingly (e.g., page 2, line 25; page 9, line 16). We also added references and discussion on previous studies (e.g., page 8, lines 8-23). To clearly demonstrate the main idea of our manuscript, we have now changed the title into “An integrated chromatin accessibility and transcriptome landscape of human pre-implantation embryos”.

2) A new assay is developed here for use in human embryo developmental studies. There is minimal validation of the assay. It should be used in human embryonic stem cells, other mammalian embryos (mouse for example) or comparable systems. The assay should have positive and negative controls that might include genetically-modified

(null) mutations that can be used to demonstrate reliability in small cell numbers (reflecting larger cell populations).

Answer: To validate our methods, we have now performed LiCAT-seq in other cell types, including human embryonic stem cells and differentiated cells, as well as the mouse early embryos. For the validation in human cells (Supplementary Fig. 1), the correlation analyses of chromatin accessibility and gene expression profiles support the high reproducibility of our methods. In addition, a comparison of both chromatin accessibility and gene expression dynamics between pluripotent and differentiated cells showed that our method can reproduce the known conclusions. One example is the reduction of expression and binding of pluripotency transcription factors (OCT4 and NANOG). Likewise, the robustness of our methods is also supported by the profiles of mouse embryos (Supplementary Fig. 2). In addition to the new figures, we have now added more description in the main text (page 3, lines 12-30; page 4, lines 1-12). The summary statistics of the new datasets have been included in the Supplementary Table 2.

3) The parameters of LiCAT-seq that enable use of the assay with small cell numbers (relative to traditional assays) should be described in more detail. What enabled use of this assay when others have failed?

Answer: We have now added more description in the main text (page 3, lines 4-12). Briefly, for chromatin accessibility profiling, the major experimental improvements include: (1) complete lysis of nuclei after the Tn5 tagmentation step; and (2) purification of genomic DNA after pre-amplification using primers targeting Tn5 adaptors. These steps can enhance the yield of DNA during library preparation using low-input materials.

4) The number of oocytes and embryos used must be included, along with the history (from the same woman, couple, or not; general morphology or indicators of viability; reason for donation (family has already been established, supernumerary, etc)).

Answer: We agree and have now added more detailed information about the embryos used in this study (Supplementary Table 1 and Supplementary Methods).

5) The table describing blastomere number (supplementary table 1) does not make much sense. One cannot collect 10 blastomeres from an oocyte, for example. The table should be clarified.

Answer: We agree and have now modified the table (Supplementary Table 1).

On page 10 of the manuscript, the essence of the work is described by the authors: “In summary, we performed a genome-wide survey of accessible chromatin.....” “We revealed strong associations....” This is a research paper that describes a survey

method that has uncovered associations. Overall, the data is not proving the title of the manuscript but suggests the implications of the title. If the issues above were addressed, it might be more suitable for publication.

Answer: We thank the reviewer for the great questions and suggestions, which help improved the quality of our manuscript. We hope that the reviewers will be satisfied with our answers.

REFERENCES

1. Guo H, *et al.* DNA methylation and chromatin accessibility profiling of mouse and human fetal germ cells. *Cell research* **27**, 165 (2017).
2. Carone BR, *et al.* High-resolution mapping of chromatin packaging in mouse embryonic stem cells and sperm. *Developmental cell* **30**, 11-22 (2014).
3. Zhu P, *et al.* Single-cell DNA methylome sequencing of human preimplantation embryos. *Nature genetics* **50**, 12 (2018).
4. Zhang Y, *et al.* Model-based analysis of ChIP-Seq (MACS). *Genome biology* **9**, R137 (2008).
5. Wang J, *et al.* Primate-specific endogenous retrovirus-driven transcription defines naive-like stem cells. *Nature* **516**, 405 (2014).
6. Grow EJ, *et al.* Intrinsic retroviral reactivation in human preimplantation embryos and pluripotent cells. *Nature* **522**, 221 (2015).
7. Gu T-P, *et al.* The role of Tet3 DNA dioxygenase in epigenetic reprogramming by oocytes. *Nature* **477**, 606 (2011).
8. Gao L, *et al.* Chromatin Accessibility Landscape in Human Early Embryos and Its Association with Evolution. *Cell* **173**, 248-259 e215 (2018).
9. Wu J, *et al.* Chromatin analysis in human early development reveals epigenetic transition during ZGA. *Nature* **557**, 256 (2018).

Reviewer #1 (Remarks to the Author):

I congratulate the authors for their efforts towards answering my criticisms and providing interesting additional data.

Reviewer #2 (Remarks to the Author):

The authors have addressed my major concerns.

Reviewer #3 (Remarks to the Author):

The authors have been responsive to the comments that I presented. In addition, I believe that the manuscript is valuable and is not a repeat or simple extension of previous studies - it both validates and provides new insights. It is most appropriate for publication in Nature Comm in my assessment.

December 7th, 2018

Dear Editor,

We would like to thank you and the reviewers for the careful assessment of our work and the comments that have helped us improve the quality of our manuscript. Here we provide a point-by-point answer to the reviewers below.

Best wishes,

Ge Lin on behalf of all authors

Reviewers' comments:

Reviewer #1 (Remarks to the Author):

I congratulate the authors for their efforts towards answering my criticisms and providing interesting additional data.

Answer: We thank the reviewer for supporting the publication of our study.

Reviewer #2 (Remarks to the Author):

The authors have addressed my major concerns.

Answer: We thank the reviewer for supporting the publication of our study.

Reviewer #3 (Remarks to the Author):

The authors have been responsive to the comments that I presented. In addition, I believe that the manuscript is valuable and is not a repeat or simple extension of previous studies - it both validates and provides new insights. It is most appropriate for publication in Nature Comm in my assessment.

Answer: We thank the reviewer for supporting the publication of our study.